# PROTEX: A RETRIEVAL-AUGMENTED APPROACH FOR PROTEIN FUNCTION PREDICTION

## ABSTRACT

Mapping a protein sequence to its underlying biological function is a critical problem of increasing importance in biology. In this work, we propose ProtEx, a retrieval-augmented approach for protein function prediction that leverages exemplars from a database to improve accuracy and robustness and enable generalization to unseen classes. Our approach relies on a novel multi-sequence pretraining task, and a fine-tuning strategy that effectively conditions predictions on retrieved exemplars. Our method achieves state-of-the-art results across multiple datasets and settings for predicting Enzyme Commission (EC) numbers, Gene Ontology (GO) terms, and Pfam families. Our ablations and analysis highlight the impact of conditioning predictions on exemplar sequences, especially for classes and sequences less well represented in the training data.

## 1 INTRODUCTION

Proteins perform a wide array of functions within organisms, captured by biologists using ontologies such as Gene Ontology (GO) terms (Ashburner et al., 2000), Enzyme Commission (EC) numbers (Tipton & Boyce, 2000), and Pfam families (Bateman et al., 2004). Mapping proteins to such functional annotations can address key problems in biology, medicine, and chemistry (Price et al., 2018; Durairaj et al., 2023). However, given the expense of wet lab experiments, and the rapid growth of protein sequence databases (Uniprot, 2023), it is critical to extend coverage using computational protein function predictions. Protein function prediction techniques largely fall into two categories, with different strengths and weaknesses. Homology-based approaches align the query protein to annotated sequences via methods such as BLAST (Altschul et al., 1997) or profile hidden Markov models (Eddy, 1998) allowing propagation of a label to the query. More recently, deep learning approaches directly predict a protein's function from its amino acid sequence.

While these approaches have been successful, key obstacles remain. A critical challenge is out-of-distribution generalization across both label and sequence space. For example, recent expansions of the Pfam database were driven by the identification of new classes (Mistry et al., 2021), for which there are few annotated sequences. Moreover, a substantial number of proteins belong to the "dark matter" of the protein universe (Durairaj et al., 2023), *i.e.* their sequences are dissimilar from those of any characterized proteins, posing considerable challenges for both homology-based and deep learning methods.

In this work, we propose ProtEx, a method for Protein function prediction via retrieved Exemplars. ProtEx is a semiparametric approach that combines aspects of non-parametric similarity search based methods and parametric deep learning models to achieve increased accuracy and robustness. ProtEx is inspired by retrieval augmented methods in natural language processing and vision, *e.g.* (Lewis et al., 2020; Pasupat et al., 2021; Izacard et al., 2023; Yu et al., 2023a; Long et al., 2022), which show advantages over fully parametric models in capturing tail information (Kandpal et al., 2023) and performing few-shot tasks through conditioning on task exemplars (Min et al., 2022; Chen et al., 2022).

As shown in Figure 1, given a query protein and candidate label, ProtEx first uses a homology-based retriever (such as BLAST) to obtain a class-conditioned set of positive and negative exemplars from the training set. Our neural model is trained to make a task-based comparison across the set of exemplars and the query to output a binary decision of whether the query sequence has the same label as the positive exemplars. To enable the model to efficiently learn the relationship between

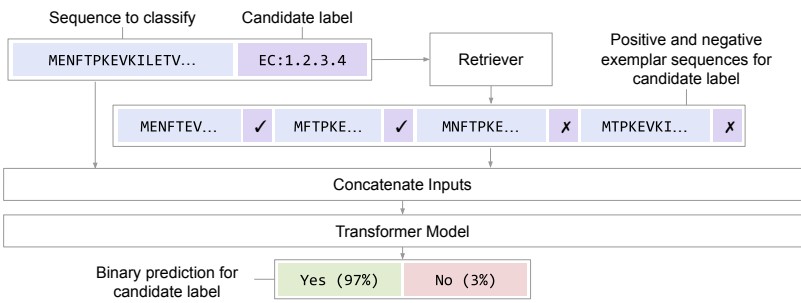

Figure 1: **Method Overview**. Our proposed method, ProtEx, predicts the relevance of various functional annotations for a given query sequence. First, for a given candidate label, positive and negative exemplar sequences are retrieved from the training data. Second, a pre-trained neural model jointly conditions on the query sequence and retrieved exemplars to make a prediction.

query and exemplar sequences, we propose a novel multi-sequence pretraining objective (Figure 3). Our formulation allows us to effectively generalize to rare and unseen classes at test time and better classify sequences that have low sequence similarity with the training data.

We evaluate our approach, ProtEx, on EC number, GO term, and Pfam family prediction tasks, showing that it achieves state-of-the-art performance across multiple settings, and consistently outperforms both existing traditional homology-based approaches and deep learning models. Our stratified analysis shows that ProtEx brings most notable improvements on rare classes and sequences that are far from the training set. We also demonstrate generalization to labels not seen at training time. Finally, our ablations and analyses highlight the efficacy of our pretraining strategy and the model's ability to leverage exemplar sequences for improved classification accuracy.[1]

## 2 BACKGROUND AND RELATED WORK

We review traditional methods for protein similarity search, neural models for protein function prediction, and related work on retrieval-augmented models from other domains.

**Protein Similarity Search**   Proteins with similar sequences often perform similar functions, and for a given query sequence, *homologous* sequences can be retrieved from a database using a variety of similarity-based search methods (Altschul et al., 1997; Johnson et al., 2010; Remmert et al., 2012). These enable *homology-based inference* (Loewenstein et al., 2009), where functional labels from retrieved homologs are transferred to the query. ProtEx can be seen as performing homology-based inference where information from the homologs is aggregated using a learned, non-linear model. A ubiquitous tool for similarity search is BLAST (Altschul et al., 1990; 1997), which identifies and scores local sequence alignments using an empirically-derived substitution cost matrix (Henikoff & Henikoff, 1992). Our method leverages BLAST and BLAST-inspired techniques to retrieve exemplars (see §4). BLAST also serves as a strong baseline for homology-based inference when used in isolation.

**Neural Models for Protein Function Prediction**   Recent work has shown that deep models mapping a protein sequence to functional predictions can outperform traditional alignment-based techniques. Models can be fit from scratch (Kulmanov et al., 2018; Ryu et al., 2019; Cao & Shen, 2021; Bileschi et al., 2022; Fan et al., 2022; Sanderson et al., 2023) or fine-tuned from a model pretrained on unlabeled protein sequences (Strodthoff et al., 2020; Dohan et al., 2021; Villegas-Morcillo et al., 2021; Yuan et al., 2023; Dickson & Mofrad, 2023). The amino acid sequence largely specifies a protein's structure and function (Anfinsen, 1973), and hence is often used as input to models. However, other approaches also encode protein structures (Sokolov & Ben-Hur, 2010; Roy et al., 2012; Konc et al., 2013; Gligorijević et al., 2021; Zhang et al., 2022; Lai & Xu, 2022), with broad coverage due to advancements in protein structure prediction (Jumper et al., 2021; Baek et al., 2021). Using structures

---

[1]Code and model predictions are available at http://anonymized.

in addition to or instead of protein sequences, either for prediction or retrieval, is largely orthogonal to the contributions of this paper, and future extensions to $\mathrm{ProtEx}$ could incorporate such techniques. Disordered proteins, which are not well characterized by a static structure would need to be handled carefully, however (Ruff & Pappu, 2021). We do not consider prediction approaches based on protein interaction networks (Mostafavi et al., 2008; You et al., 2019; Zhang et al., 2019; Kulmanov et al., 2024), which have demonstrated high accuracy where such networks are already established, but have limited overall coverage.

Particularly relevant are recent methods combining neural models and retrieval for protein function prediction. Notin et al. (2022) combines likelihood scores from a protein language model and an MSA to predict the functional effects of mutations. Hamamsy et al. (2023) distills a protein similarity function into a neural network, and Dickson & Mofrad (2023) fine-tunes a retrieval model for homology-based inference. The ProtIR method of Zhang et al. (2024) uses transductive learning where predictions from a retriever and neural classifier are iteratively updated towards agreement. In contrast to retrieval methods, which form predictions by aggregating pairwise similarity scores over sequences, $\mathrm{ProtEx}$ predicts conditioned on multiple labeled exemplar sequences. This also differs from methods that use the correlation structure in aligned sets of homologous sequences (MSAs), without accompanying annotations of their function, to inform, *e.g.*, structure prediction (Marks et al., 2011; Jumper et al., 2021; Baek et al., 2021) or representation learning (Rao et al., 2021).

**Retrieval-Augmented Models and In-Context Fine-tuning** Retrieval augmented neural models have shown success for a variety of text generation and classification tasks, with particular strength in recalling long tail knowledge, where even large parametric language models (LMs) struggle (Kandpal et al., 2023). Examples include retrieval-augmented language models, and models for machine translation, question answering, semantic parsing, and text classification (Khandelwal et al., 2020; 2021; Karpukhin et al., 2020; Guu et al., 2020; Lewis et al., 2020; Izacard et al., 2023; Pasupat et al., 2021; Yu et al., 2023a; Chalkidis & Kementchedjhieva, 2023). Retrieved context or exemplars come from large-scale unlabeled collections or labeled sets for supervised learning (Wang et al., 2022; Lewis et al., 2021). Computer vision and multimodal models also benefit from retrieval augmentation (Long et al., 2022; Ramos et al., 2023). A focus in NLP have been methods to adapt a pretrained parametric language model into one that can integrate information from retrieved sequences, and joint training of a retriever and generation model. In this work, we use pre-existing protein similarity models for retrieval and pre-train a protein sequence model to make inferences from multiple proteins.

$\mathrm{ProtEx}$ is also related to in-context tuning methods for few-shot tasks (Min et al., 2022; Chen et al., 2022), where pretrained language models are meta-trained to make predictions given an input and task-relevant exemplars. These works show strong performance on unseen tasks, enabled by the LM's ability to make predictions from an input and a few in-context exemplars. Similarly, $\mathrm{ProtEx}$ can be seen as meta-training on multiple binary classification tasks with in-context exemplars and, as shown in Section 4.5, can adapt to new labels without additional fine-tuning.

## 3 PROPOSED METHOD

Given a protein represented as a sequence of amino acids, $x$, our goal is to predict a set of associated labels, $y \subset \mathcal{L}$, where $\mathcal{L}$ is a set of protein function labels, such as EC numbers, GO terms, or Pfam labels. Core to our approach is to condition model predictions on a set of annotated exemplar sequences, which are retrieved from a training set, as visualized in Figure 2.

### 3.1 OVERVIEW

To predict a set of labels $y$ given $x$, we first determine a set of candidate labels, $\hat{\mathcal{L}}_x \subseteq \mathcal{L}$, and then make an independent binary prediction for each candidate label $l \in \hat{\mathcal{L}}_x$, conditioning on exemplars that are selected based on both $x$ and $l$. Due to our model's limited context window, we can only condition on a limited number of exemplars per prediction. Therefore, our approach has several advantages compared to generating $y$ based on $x$ and a single limited set of exemplars. First, we can focus each prediction on the exemplars that are most relevant towards understanding the class boundary of a specific label $l$. Second, we can consider a larger and more diverse set of candidate labels than those corresponding to a limited number of exemplars. Finally, our approach can be seen

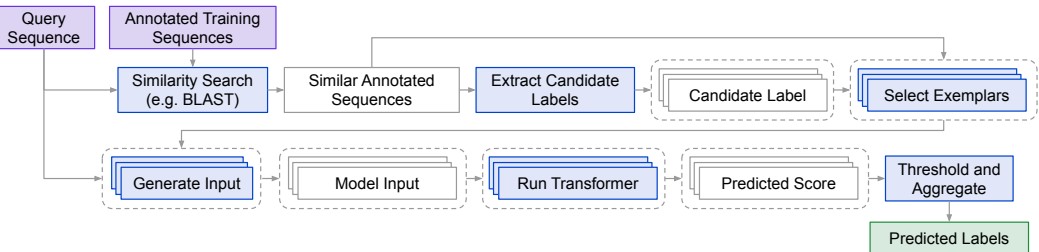

Figure 2: **Inference Procedure**. Overview of procedure for computing a set of predicted labels given a protein sequence and a set of training sequences with annotated function.

as meta-training on multiple binary classification tasks with in-context exemplars, which we show enables ProtEx to adapt to new labels without additional fine-tuning.[2]

The inference procedure for predicting a set of functional labels, $y$, given a protein sequence, $x$, is shown in Figure 2. We review each step of this procedure below.

**Similarity Search** Given a query sequence, $x$, the first step in Figure 2 is to retrieve a ranked list of relevant sequences with known functional annotations, denoted $\mathcal{N}_x$, from the training set. We adopt standard methods based on efficiently computing local alignments between sequences. We use BLAST (Altschul et al., 1997) for most experiments, which has shown strong performance and is computationally feasible to run on most of the datasets we study.[3] Specifically, we run `blastp` to retrieve up to the 100 most similar sequences in the training set, and rank these sequences based on the BLAST alignment score. (See Appendix B.5 for details.)

**Candidate Labels** We determine the set of candidate labels, $\hat{\mathcal{L}}_x$, as the union of the labels corresponding to sequences in $\mathcal{N}_x$. This simple approach ensures that we have at least one positive exemplar sequence for every candidate label.

**Selecting Exemplars** We select a set of positive exemplars, $\mathcal{E}_{x,l}^p$, as the top-$k^p$ sequences in $\mathcal{N}_x$ that are annotated with the candidate label $l$. Similarly, we select a set of negative exemplars, $\mathcal{E}_{x,l}^n$, as the top-$k^n$ sequences in $\mathcal{N}_x$ that are *not* annotated with $l$. The hyperparameters $k^p$ and $k^n$ are discussed in Section 4 and selected such that the exemplars fit within the model's context window.

**Model Input** As visualized in Figure 1, we form a retrieval-augmented input sequence by concatenating the query sequence, $x$, the candidate label $l$, and the positive and negative exemplars, $\mathcal{E}_{x,l}^p$ and $\mathcal{E}_{x,l}^n$, with a character denoting whether each exemplar is positive or negative.[4]

**Transformer Model** As the focus of our study is the pre-training and fine-tuning recipe for our model, we use the general encoder-decoder Transformer (Vaswani et al., 2017) architecture of T5 (Raffel et al., 2020). As demonstrated in Raffel et al. (2020), this allows for flexibility in defining different pre-training and fine-tuning tasks without changes to the underlying architecture. We evaluate Small (60M) and Base (220M) sized models, using the Base size for our main results. We also evaluated Transformer variants that offer more efficient handling of long context. See Appendix C.2 for these comparisons. Similarly to Raffel et al. (2020), to apply our encoder-decoder model as a binary classifier, our model predicts single character sequences corresponding to either positive (p) or negative (n) predictions. We define the model score $s_\theta(x, l, \mathcal{E}_{x,l}^p, \mathcal{E}_{x,l}^n)$ with trainable parameters $\theta$ as the probability the model assigns to the positive character sequence p, i.e. $s_\theta(x, l, \mathcal{E}_{x,l}^p, \mathcal{E}_{x,l}^n) := \log P_\theta(\text{p}|x, l, \mathcal{E}_{x,l}^p, \mathcal{E}_{x,l}^n)$. Since we consider separate exemplars for each candidate label, this requires running the model up to $|\hat{\mathcal{L}}_x|$ (the number of candidate labels) times for each input sequence. We analyze the computational cost of inference in Appendix B.3.

---

[2]While such a capability is theoretically also possible if generating a set of labels, e.g., by anonymizing labels (Pasupat et al., 2021), this would be conceptually less straightforward.

[3]For the Pfam task, which requires retrieval from a larger training set, we use a similar but alternative approach discussed in Section 4.4.

[4]See Appendix A.1 for further details. We also ablate inclusion of the candidate label in Section 4.5.

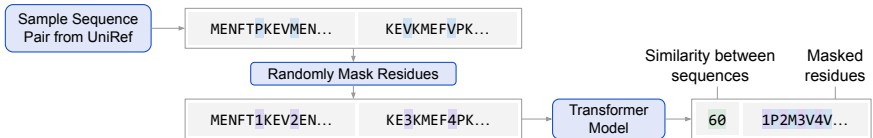

Figure 3: **Pretraining task.** We sample a pair of unlabeled sequences, and the model is tasked with predicting masked residues and a similarity score between the two sequences.

**Aggregating Predictions** For multilabel prediction tasks (e.g. EC and GO), we can determine a predicted label set, $\hat{y}$, based on a score threshold $t$:

$$\hat{y} = \{l \mid l \in \hat{\mathcal{L}}_x \wedge s_\theta(x, l, \mathcal{E}^p_{x,l}, \mathcal{E}^n_{x,l}) > t\},$$

and varying $t$ produces a tradeoff between precision and recall. For multiclass prediction tasks (e.g. Pfam) where $y$ is always a singleton, we can determine the class with the highest score:

$$\hat{y} = \{\operatorname*{arg\,max}_{l \in \hat{\mathcal{L}}_x} s_\theta(x, l, \mathcal{E}^p_{x,l}, \mathcal{E}^n_{x,l})\}.$$

### 3.2 TRAINING

Model training consists of two stages. First, we pre-train using unlabeled sequence pairs. Then, we fine-tune for a specific task by constructing positive and negative examples given a set of labeled sequences. In both stages, models are trained to maximize the likelihood of a target sequence given an input sequence, given our generic encoder-decoder architecture (Raffel et al., 2020).

**Pre-training** Our pre-training task shown in Figure 3. Prior work has primarily pre-trained on single unlabeled sequences, then fine-tuned with a single sequence as input to improve protein function prediction (Dohan et al., 2021; Lin et al., 2023). In contrast, our goal is to train models that implicitly compare a query sequence with exemplar sequences. We thus propose a new pre-training objective over *multiple sequences*. We sample pairs of sequences from UniRef90 (Suzek et al., 2015), approximating a uniform distribution over sequence similarity buckets. We then implement a version of the span denoising objective of Raffel et al. (2020). We mask the sequences by randomly replacing approximately 10% of residues with a placeholder index. The prediction target consists of predicting the masked residues as well as the similarity score between the two sequences. As a simple and easy to compute measure of sequence similarity, we use Levenshtein distance normalized by the sequence length, bucketed to the nearest multiple of 5. Intuitively, the prediction target encourages the model to learn to approximately align and compare the two input sequences. We then expect the model to implicitly learn a more task-specific notion of similarity during fine-tuning. Pre-trained checkpoints are shared across all tasks that we study. See Appendix A.2 for details and §4 for analysis.

**Fine-tuning** For each sequence $x$ and corresponding set of annotated labels $y \subset \mathcal{L}$ in the training set, we create both positive and negative examples, with targets p and n, respectively. We generate a positive example for every label $\in y$, and negative examples corresponding to labels $\notin y$.[5] We determine the retrieval-augmented input sequence as described in §3.1. The only difference is how we select positive and negative exemplars, $\mathcal{E}^p_{x,l}$ and $\mathcal{E}^n_{x,l}$, from the ranked list of related sequences, $\mathcal{N}_x$.[6] We evaluated top-k (as used at inference time), uniform, and geometric sampling (Pasupat et al., 2021).[7] Uniform or geometric sampling leads to greater diversity in the training data compared to training for multiple epochs over data generated using deterministic top-$k$ selection. Also, for cases where the training and evaluation sequences are not from the same distribution, such sampling can better align the distribution between query sequences and exemplar sequences seen at training time with the distribution seen at inference time. More analysis is given in §4.5.

---

[5]See Appendix B.6 for details of how negative examples are sampled, and a discussion of class imbalance.

[6]At training time we also need to ensure the query sequence $x$ is excluded from the ranked list of related sequences.

[7]Geometric sampling samples the $j$th element with probability $\propto p(1-p)^{(j-1)}$, where parameter $p$ provides interpolation between top-$k$ and uniform sampling.

## 4 EXPERIMENTS AND ANALYSIS

We compare the performance of PROTEX with other approaches (§4.2) across several EC, GO, and Pfam classification tasks (§4.1), with results reported in in §4.3 and §4.4. Finally, in §4.5, we report additional analysis and ablations. Further details and results can be found in Appendices B and C.

### 4.1 TASKS AND DATASETS

Table 1: **Dataset Statistics**. We consider prediction of EC numbers, GO terms, and Pfam families across several settings proposed in prior work. We report the number of unique classes among the training sequences, and the average number of classes per sequence.

| Name | Labels | Training Sequences | Classes | Avg. # Classes Per Seq. |
|---|---|---|---|---|
| Random EC | EC | 438,522 | 4,862 | 1.89 |
| Random GO | GO | 438,522 | 31,365 | 45.49 |
| Clustered EC | EC | 182,965 | 3,411 | 1.90 |
| Clustered GO | GO | 182,965 | 26,538 | 45.57 |
| NEW-392 | EC | 227,362 | 5,242 | 1.06 |
| Price-149 | EC | 227,362 | 5,242 | 1.06 |
| PDB EC | EC | 15,551 | 538 | 1.67 |
| Clustered Pfam | Pfam | 1,296,280 | 17,929 | 1 |

We consider several EC number, GO term, and Pfam prediction tasks summarized in Table 1. Some additional details are provided in Appendix B.

For EC and GO prediction, we adopt the random and clustered splits from Sanderson et al. (2023). These splits consist of proteins and their corresponding annotations from Swiss-Prot, the manually curated portion of UniProt (Consortium, 2015). For the random split, approximately 80% of proteins were reserved for training, with 10% each assigned to the development and test sets. For the more challenging clustered split, proteins were divided evenly into train, development, and test sets based on UniRef50 clusters (Suzek et al., 2015), such that proteins in the development and test sets have lower sequence similarity with those in the training set than a random split.

To better compare our results on EC prediction with prior work, we also consider several other evaluations. First, we evaluate EC prediction on the setting proposed in Yu et al. (2023b). Similarly to Sanderson et al. (2023), this setting uses sequences from Swiss-Prot for training. There are two evaluation sets, NEW-392 and Price-149. While the training data contains sequences added to Swiss-Prot prior to April 2022, New-392 consists of 392 enzyme sequences added after this date, forming a temporal split. Price-149, originally curated by Sanderson et al. (2023), consists of 149 sequences with EC numbers determined experimentally by Price et al. (2018). These sequences were inconsistently labeled by automated annotation methods, indicating a challenging setting. Second, we also adopt the PDB-based dataset proposed by Gligorijević et al. (2021). In contrast to the other EC splits based on sequences from Swiss-Prot, this dataset focuses on proteins with experimentally determined structures in PDB (Berman et al., 2000). Therefore, the training set is considerably smaller. Notably, our method does not require structural information, but we nevertheless evaluate on this setting to compare with prior work.

Finally, we also evaluate on the Pfam seed dataset (Finn et al., 2014) where the goal is to map each protein sequence to one of 17,929 families. We use the clustered split as defined by Bileschi et al. (2022) where sequences in the development and test sets have less than 25% sequence identity to those in the training set.

### 4.2 BASELINES

For each task we consider three types of baselines. First, we consider the strongest performing neural models and other methods from prior work. Second, to determine the impact of retrieving and conditioning on exemplars, we evaluate PROTEX when no exemplars are included during fine-tuning and inference. Third, we report results for BLAST, following the setting of Sanderson et al. (2023), which imputes the labels from the most similar sequence returned by BLAST if the score is above

Table 2: Max F1 scores for EC and GO prediction on the random and clustered SwissProt-based splits proposed by Sanderson et al. (2023).

| | EC | | GO | |
|---|---|---|---|---|
| Method | Random | Clustered | Random | Clustered |
| ProteInfer | 0.977 | 0.914 | 0.885 | 0.782 |
| ProteInfer (ensemble) | 0.981 | 0.930 | 0.899 | 0.811 |
| BLAST | 0.984 | 0.950 | 0.902 | 0.824 |
| ProtEx | **0.987** | **0.958** | **0.917** | **0.854** |
| ProtEx (no exemplars) | 0.977 | 0.944 | 0.786 | 0.754 |

an alignment score threshold. In some cases, our BLAST results are stronger than those reported by prior work due to this thresholding. Notably, this BLAST baseline frequently outperforms the strongest neural methods from prior work, highlighting the importance of including such baselines, as well as motivating methods such as ProtEx that can combine the strengths of similarity-search methods such as BLAST with pre-trained neural models.

## 4.3 EC AND GO MAIN RESULTS

Here we report results on several EC and GO classification tasks. We used up to 2 positive and 2 negative exemplars. See Appendix B for further details and Appendix C.1 for the statistical significance of key comparisons (tested differences are significant with $p$-values $< 0.01$).

First, we report results in Table 2 for the random and clustered EC and GO splits proposed by Sanderson et al. (2023). We follow Sanderson et al. (2023) and report the maximum micro-averaged F1 score. We compare with a single-model and ensembled versions of ProteInfer (Sanderson et al., 2023), a CNN-based model. Our BLAST result reproduces that of Sanderson et al. (2023), which outperforms ProteInfer. ProtEx also improves over BLAST, with performance dropping considerably if exemplars are not included in the input during fine-tuning and inference. The precision and recall curves of ProtEx relative to BLAST are shown in Figure 4, demonstrating improvements in precision at all recall values.

Second, we report in Table 3 results on the NEW-392 and Price-149 evaluations proposed by Yu et al. (2023b), and compare with their proposed method, CLEAN. As there is no development set provided in this setting, we used the hyperparameters determined for the clustered EC task, and compare

Table 3: Weighted AUC for EC prediction for NEW-392 and Price-149.

| Method | NEW-392 | Price-149 |
|---|---|---|
| CLEAN | 0.740 | 0.733 |
| BLAST | 0.788 | 0.691 |
| ProtEx | **0.932** | **0.842** |
| ProtEx (no ex.) | 0.926 | 0.839 |

Table 4: Max protein-centric F1 for EC prediction on PDB-based split.

| | PDB EC | | |
|---|---|---|---|
| Method | 30% | 50% | 95% |
| DeepFRI | 0.470 | 0.545 | 0.631 |
| ESM-1b | 0.737 | 0.797 | 0.864 |
| GearNet MVC | 0.744 | 0.808 | 0.874 |
| ESM-GearNet | — | — | 0.890 |
| PromptProtein | 0.765 | 0.823 | 0.888 |
| ProtST (ESM-2) | — | — | 0.878 |
| ESM-2 (adapter) | — | — | 0.892 |
| PST (ESM-2) | — | — | 0.899 |
| BLAST | 0.801 | 0.848 | 0.900 |
| ProtEx | **0.820** | **0.862** | **0.909** |
| ProtEx (no ex.) | 0.717 | 0.777 | 0.849 |

results based on the weighted AUC metric reported by Yu et al. (2023b), which does not require selecting a score threshold. We see that ProtEx improves over both CLEAN and BLAST. Again, ablating exemplars leads to a drop in performance. Additionally, we report F1 results in Appendix C.3.3.

Finally, we also report results on the PDB-based split proposed by Gligorijević et al. (2021) in Table 4. Following prior work, we report the maximum protein-centric F1 score (i.e. $F_{max}$), and stratify results by the maximum similarity between test sequences and training sequences. Table 4 includes results for the strongest methods from prior work. We report results for DeepFRI from Gligorijević et al. (2021), ESM-1b (Rives et al., 2021) and GearNet from Zhang et al. (2022), ESM-GearNet from Zhang et al. (2023), PromptProtein from Wang et al. (2023), ProtST from Xu et al. (2023), and

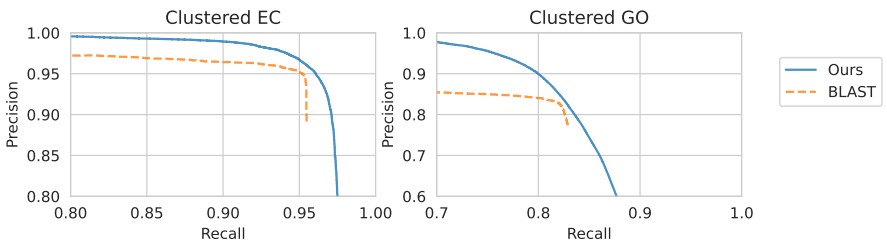

Figure 4: Precision and recall of ProtEx and BLAST on the clustered EC and GO tasks.

ESM-2 (Lin et al., 2023) with a classifier head and PST from Chen et al. (2024). Perhaps surprisingly, we see that BLAST with an alignment score threshold achieves a very competitive result on this setting. Regardless, ProtEx outperforms BLAST and other previously reported results on this setting, including ESM-2 (a 15B parameter model), other models based on ESM-2 (ESM-GearNet, ProtST, and PST), and approaches that explicitly consider structural information. Ablating exemplars again demonstrates the significance of conditioning on exemplars. For future work, improvements from pre-training scale (e.g. ESM-2) and incorporating structural information and other resources (e.g., GearNet, ProtST, PST, PromptProtein) could be complementary to our retrieval-augmented approach.

### 4.4 PFAM MAIN RESULTS

For the Pfam dataset, we use an alternative to BLAST for selecting similar sequences, detailed in Appendix B.5. The Pfam dataset is considerably larger than the other datasets (see Table 1), and running BLAST for all examples in the training set can take considerable time. Moreover, BLAST does not achieve as strong of a result for Pfam classification as it does for the EC and GO tasks. Therefore, we implemented an alternative retrieval system that can be more easily parallelized and customized than BLAST. For every sequence, we select a set of similar sequences for each class independently. For efficiency, we randomly select up to a maximum number of sequences per class in the training set, and then rank these sequences according to a local alignment score that is similar to the one computed by BLAST. Based on analysis of the effect of restricting the number of classes (see Appendix C.4.1), we opted to consider all classes as candidate labels. We also evaluated this strategy for EC prediction although it did not perform as well as our BLAST-based approach (see Appendix C.3.2). We use 4 positive exemplars and zero negative exemplars, since we found in our early experiments that additional positive exemplars added more benefit.

Table 5: **Results on the Pfam clustered split.** Sequences in the test set have less than 25% sequence identity to the training set.

| Method | Family Accuracy | Lifted Clan Accuracy | Avg. Per-Family Accuracy |
|---|---|---|---|
| Top pick HMM | 81.9 | 88.1 | 82.9 |
| BLAST | 64.1 | 70.1 | 63.7 |
| ProtENN | 87.8 | 89.0 | 80.4 |
| ProtNLM | 87.4 | 90.7 | 80.6 |
| ProtTNN | 88.4 | 90.5 | 83.4 |
| ProtTNN (ensemble) | 89.7 | 91.7 | 85.0 |
| ProtEx | **92.6** | **93.3** | **91.7** |
| ProtEx (no exemplars) | 76.3 | 80.2 | 65.7 |

We compare ProtEx with two strong homology-based approaches (BLAST and Top pick HMM, as described in Dohan et al. (2021)), ProtENN (Bileschi et al., 2022) a convolutional neural network ensemble, and ProtNLM (Gane et al., 2022) and ProtTNN (Dohan et al., 2021), which are pretrained Transformer models. For the no exemplar ablation, we found that predicting a binary label without exemplars generalizes poorly when the number of classes is large, and so we instead finetune our pretrained checkpoint to predict the class label as a string given the sequence, which we found performed better. See Appendix § C.4.2 for further discussion.

Results are shown in Table 5. We report family accuracy, lifted clan accuracy that groups families into higher level clans (Dohan et al., 2021), and the average per-family accuracy, which gives equal weight to all classes, including rare classes. Our approach achieves state-of-the-art performance by a considerable margin. Additionally, as shown in Figure 5, while other methods typically show lower performance for examples with rare labels, ProtEx demonstrates more consistent performance across training set family sizes, showcasing large improvements for examples belonging to rare families. See Appendix C.4.3 for further stratified performance analysis, showing that our method performs well across sequences that have low similarity with the closest sequence in the training data, and that similar trends are observed at the lifted clan level.

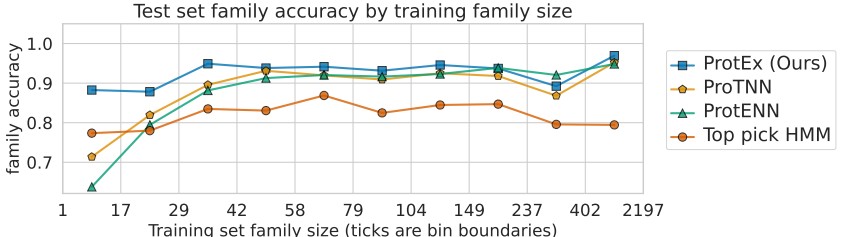

Figure 5: **Pfam stratified family accuracy**. ProtEx shows considerably improved performance for sequences belonging to rare labels.

## 4.5 ANALYSIS AND ABLATIONS

Table 6: **Generalization to Unseen Labels.** Max F1 on clustered EC and GO when a randomly selected subset of labels are not included during fine-tuning.

| | EC | | GO | |
|---|---|---|---|---|
| Method | Seen | Unseen | Seen | Unseen |
| BLAST | 0.953 | 0.964 | 0.826 | 0.816 |
| ProtEx | **0.960** | **0.970** | **0.849** | **0.839** |

Table 7: **Pre-training Ablations.** We report F1 on Clustered EC development split for different pre-training strategies.

| Pre-training | F1 |
|---|---|
| Sequence pair w/ score | **0.958** |
| Sequence pair | 0.956 |
| Single sequence | 0.952 |
| No pre-training | 0.912 |

**Generalization to New Labels**  We tested the ability of ProtEx to make predictions for new classes not seen during fine-tuning. On the EC and GO clustered splits we randomly removed 10% of classes during fine-tuning, while retaining the ability to retrieve sequences annotated with these classes at inference time. As shown in Table 6, ProtEx performs comparably or better than BLAST even on classes it has not seen during training. Relatedly, we also found that there is only a minimal performance decrease when candidate labels are not included in the model input (see Appendix C.3.1), further indicating that the model is conditioning its predictions on the exemplar sequences as opposed to directly representing the sequence to class relationship in the model parameters.

**Pre-training Analysis**  We show results for ablating the key elements of our pre-training recipe in Table 7. Notably, there is a drop in end task performance when pre-training with only a single sequence as input, as commonly done in prior work, as opposed to pre-training over sequence pairs. This indicates that our pre-training task is useful for retrieval-augmented models that are fine-tuned to make comparisons across multiple sequences. See Appendix B.4 for further details.

**Model Architectures and Scaling**  In Appendix C.2 we compare Small and Base sized models, finding that there is a modest benefit to increasing model size from Small to Base. We also compare the Fusion-in-Decoder (FiD) approach introduced by Izacard & Grave (2021) with a standard Transformer, finding that this may be viable path towards scaling to more exemplars, but that there is a performance drop, indicating that cross-attention between exemplar sequences is beneficial.

**Exemplar Distribution and Sampling** As described in §3, we study different sampling strategies to select exemplars during training. This adds diversity to the training data, and can also help align the distribution of similarities between query and exemplar sequences seen during training with those seen during inference, which is especially useful for non-random splits. We highlight this capability on the Pfam task, which features the largest distributional shift between training and inference due to the split restricting inference sequences to have <25% sequence similarity to the training set. Figure 6 shows the corresponding distribution of similarities between query and exemplar sequences and its impact on out-of-distribution generalization as measured by family accuracy on development set.

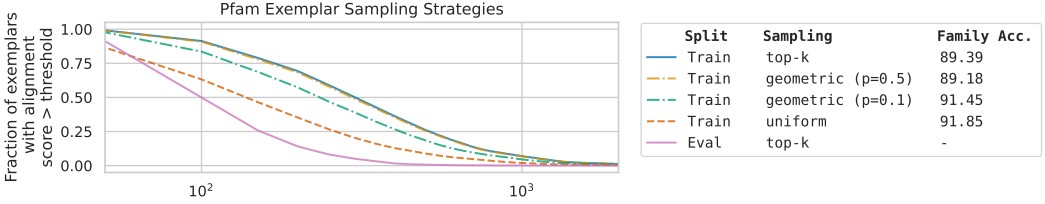

Figure 6: **Effect of exemplar sampling strategy on Pfam**. We visualize how different sampling strategies can mitigate the distribution shift between training and evaluation. Uniform sampling leads to the training distribution most similar to the evaluation distribution, and also leads to the highest family accuracy on the development set.

## 5 LIMITATIONS AND DISCUSSION

**Computational Requirements** While we use a model size and number of pre-training steps comparable to or less than prior work (Appendix B.3), the cost of inference with our method is potentially larger due to encoding multiple protein sequences and making independent predictions for each class. To mitigate these factors, we have considered the FiD architecture (Appendix C.2) and a candidate label generator (§3), which offer a path towards more efficient inference. Regardless, improvements in accuracy and robustness can justify an increase in computational cost for some applications, given the much greater cost of running wet-lab experiments to annotate protein function.

**Scope and Future Work** In this work we focused on training and inference procedures that can effectively condition predictions on retrieved exemplars, using a general-purpose Transformer architecture and standard methods for retrieval (e.g. BLAST). Going forward, our approach could potentially be further improved using enhanced similarity search techniques such as those based on protein structure (Zhang et al., 2022; Hamamsy et al., 2023; Van Kempen et al., 2024) and more specialized architectures. Finally, we focused on predicting EC, GO, and Pfam labels. Other tasks such as fitness prediction (Romero et al., 2013) or generating free-text descriptions of protein function (Gane et al., 2022; Abdine et al., 2024) could be of interest for future work.

**Broader Impact and Ethical Considerations** Our method enables more accurate and robust prediction of protein functional annotations. Any computationally derived annotation should be verified by wet lab experiments where possible, especially for critical applications. Our method extends a long history of prior work that develops such tools and the community's safeguards for how to apply them in an ethical manner applies here as well.

## 6 CONCLUSION

We proposed ProtEx, a semiparametric approach that combines aspects of homology-based similarity search with pre-trained neural models. ProtEx achieves state-of-the-art results on EC, GO, and Pfam classification tasks. Our work highlights the potential of retrieval-augmented methods for improving the accuracy and robustness of protein function prediction.

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

# A  ADDITIONAL METHOD DETAILS

## A.1  MODEL INTERFACE

We use the character-based vocabulary of Xue et al. (2022), which ensures that amino acid sequences are tokenized into their individual amino acid residues. We represent labels as short character sequences such as `EC:1.2.3.4`. We use single characters to indicate the start and end of amino acid sequences, and to indicate whether an exemplar sequence is positive or negative.

The training target is a single character sequence, `p` or `n`, for positive and negative examples, respectively. At inference time, we determine the score based on the probability assigned to the single character sequence `p`.

## A.2  PRETRAINING DETAILS

As our fine-tuned models need to make comparisons between query and exemplar sequences of varying similarities, we construct the dataset such that all similarity ranges are well represented in the pre-training data. For each pair, we sampled one sequence uniformly from UniRef90. Then, we sampled a second sequence, approximating a uniform distribution over similarity buckets. The resulting distribution of normalized Levenshtein similarities in the pretraining data is shown in Table 8.

The process is loosely analogous to some methods explored in NLP. For example, Sellam et al. (2020) pre-trained models to compute BLEU scores over pairs of strings. BLEU is a deterministic measure of string similarity. Models were then fine-tuned on human labeled data to learn a more task-specific notion of similarity. Pre-training on a context that includes *related* sequences is perhaps also analogous to the in-context pre-training method proposed by Shi et al. (2024), which includes a language modeling objective over related documents, showing this is useful relative to randomly selected documents, for various downstream tasks.

Table 8: **Distribution of similarities in pretraining data.** We report the fraction of the sequence pairs in the pretraining data for different ranges of normalized Levenshtein similarity.

| Similarity | Data % |
|---|---|
| 0-25% | 19.8 |
| 25-50% | 38.4 |
| 50-75% | 30.1 |
| 75-100% | 11.7 |

# B  ADDITIONAL DATASET AND EXPERIMENT DETAILS

## B.1  DATASET DETAILS

**EC Labels**  As Table 1 shows, the number of EC classes considered varies across tasks. This is partially due to differences in which sequences, and therefore which EC labels, are included in the training set. However, different tasks also consider different tiers of the EC hierarchy. The Swiss-Prot based random and clustered splits consider labels from all 4 levels of the EC hierarchy, the PDB EC tasks considers only levels 3 and 4, and NEW-392 and Price-149 evaluations only consider level 4. Also notably, the Price-149 labels were originally derived from Price et al. (2018). However, more recent work (Price et al., 2022) has revisited the functional annotations of some of these sequences, and should be considered for future work with this evaluation.

**Dataset Licenses**  The EC and GO tasks are adapted from Swiss-Prot, which is the human curated portion of UniProt that is released under CC BY 4.0. The PDB EC split is also available under CC BY 4.0. The Pfam task is derived from Pfam[8] and is released under the CC0 1.0 license. For pretraining

---

[8]https://interpro-documentation.readthedocs.io/en/latest/pfam.html

data we used Uniref90 Suzek et al. (2015), which is derived from UniProt (Consortium, 2015) that is released under CC BY 4.0.

## B.2 HYPERPARAMETERS

**Pre-training** We pre-trained models for 1M steps using a learning rate of 1e-3 and a batch size of 256 tokens using Adafactor (Shazeer & Stern, 2018).

**EC and GO Fine-tuning** We selected hyperparameters based on development set performance, focusing on the Swiss-Prot clustered splits. For all ProtEx models with exemplars we use a learning rate of 1e-3 with Adafactor with dropout regularization (Srivastava et al., 2014) set to 0.1. For the Swiss-Prot based splits, we trained models for 50,000 steps. For the smaller PDB based split, we trained models for 8,000 steps. Models without exemplars were trained longer, as these models took longer to reach a stable development accuracy. We trained these models for 100,000 steps for the Swiss-Prot splits and 40,000 steps for the PDB split. We used a batch size of 256 for all experiments. For the random and clustered Swiss-Prot splits we used a maximum input sequence length of 6784 tokens, which led to some truncation of exemplars in about 1% of examples during training and inference. The other tasks did not lead to inputs that exceeded this length.

**Pfam Fine-tuning** For all ProtEx models with exemplars we use a learning rate of 2e-4, batch size of 128, a maximum input length of 6,528, and dropout set to 0.1. We finetune for 200,000 steps with Adafactor, and pick the best model based on development accuracy.

The configuration for the no exemplar ablation is similar except we use a higher learning rate of 1e-3 which we found worked better in practice and a beam size of 8 since we are treating the label as a string that can be tokenized into multiple tokens. Since there are no exemplars the maximum input length could be shortened to 2,688.

## B.3 COMPUTATIONAL RESOURCES AND ANALYSIS

For training and inference we used Google Cloud TPUs (v3 and v5e) in configurations of up to 128 chips.

**Model Training** Pre-training the Base model to 1M steps took approximately 7 days on 64 TPU v3 chips. Fine-tuning the Base model took approximately 3 hours per 10K steps of fine-tuning.

**Retrieval** Training and inference requires retrieving exemplars. The expense of retrieving exemplars is comparable to systems such as AlphaFold (Jumper et al., 2020) or MSA Transformer (Rao et al., 2021), which retrieve sequences as a preprocessing step, albeit for a different purpose (to build a MSA).

For inference with BLAST, a query over the largest training split considered (438K SwissProt examples for the random EC and GO splits) achieved a throughput of >1 sequence per second, running `blastp -query` with `-num_threads 16` and `-max_target_seqs 100`, on a standard CPU workstation.

As we speculate in Section 5, embedding-based retrievers could potentially provide an even more computationally efficient way to retrieve exemplars in the future.

**Model Inference** We evaluated 60M (Small) and 220M (Base) parameter T5 models (Appendix C.2). Notably, even the Base model is considerably smaller than some prior work, such as ESM-2 (Lin et al., 2023), which has 15B parameters. As ProtEx outperforms several approaches based on ESM-2, our results suggest that retrieval-augmented models may offer the ability to achieve greater accuracy with smaller models.

Inference with ProtEx requires running the model once for each candidate label. The Base model throughput was approximately 500 sequence and candidate label pairs per second on a cluster of 64 Google Cloud TPU v3 chips. Inference can additionally be parallelized over multiple clusters or performed on a larger cluster. Model throughput could be improved by considering more efficient

architectures (as analyzed in Appendix C.2), and the number of candidate labels per sequence could be reduced by considering stronger candidate label generators.

### B.4 PRE-TRAINING ABLATIONS DETAILS

Here we provide additional details about the pre-training ablations shown in Table 7. As pre-training is computationally expensive and our Small and Base models perform similarly (see C.2), we used Small models for these comparisons. Additionally, we observed that fine-tuning performance was comparable when tuning from a checkpoint that had been pre-trained for 500K or 1M steps, indicating that most of the advantage of pre-training is accrued in the first 500K steps. Therefore, we compared models pre-training for 500K steps. Finally, given the full development set is quite large (approximately 180,000 examples), we perform this ablation on a random 10% subset.

### B.5 RETRIEVER DETAILS

**BLAST** We use `ncbi-blast-2.14.1+`. We run `makeblastdb` with `-dbtype prot`, and then query the database using `blastp` with default arguments and `-max_target_seqs 100`.

We select exemplars during training using geometric sampling with $p = 0.5$.

We consider only candidate labels associated with sequences in the retrieved set. The number of candidate labels per sequence can vary, e.g. the mean is 6.3 candidate labels per sequence for the Random EC dataset vs. 237.2 candidate labels per sequence for the Random GO task. This is influenced by the number of classes per sequence per Table 1.

**Per Class Retrieval** For flexibility and ease of parallelization, we use Biopython[9] `Align.PairwiseAligner`. We set `mode = local`, `extend_gap_score = -1.0`, `open_gap_score = -11.0`, and `substitution_matrix = BLOSUM62`.

For development and inference, for each query we find the closest 4 exemplars from each class using the pairwise aligner above. For training given the size of the dataset, for each query we sample up to 100 exemplar candidates per class and select 4 exemplars from this set using uniform sampling as detailed in § 3, which we found performs the best (Table 6).

### B.6 NEGATIVE EXAMPLE SAMPLING

As discussed in §3, we sometimes use sampling of negative examples to avoid class imbalance during training. When using the BLAST retriever this is not necessary because the set of candidate labels consists of a reasonable balance of positive and negative labels. However, when using the Per Class Retrieval method, naively generating a training example for every label would lead to an imbalance. Therefore, for each sequence, we generate a negative example for the label with the highest similarity score, and also randomly sample another negative label.

## C ADDITIONAL RESULTS AND ANALYSIS

### C.1 STATISTICAL SIGNIFICANCE AND VARIANCE

For the results reported in Table 2 and Table 4, we assessed the statistical significance of the difference in F1 score between PROTEX and BLAST, for cases where the difference was less than 0.01, using a permutation test Yeh (2000). In Table 9, we report the $p$-values and standard deviation of the sampled score differences under the null hypothesis that predictions from the two approaches are interchangeable, which was estimated using 100 sampled permutations of the predictions. We computed $p$-values using a t-test.

For the evaluation settings where the training and evaluation sets are small, we also computed the variance from different fine-tuning runs. The standard deviation across 3 different fine-tuning runs is shown in Table 10, which is in all cases small relative to the performance differences between PROTEX and prior work.

---

[9]https://biopython.org/

Table 9: Statistical significance of comparisons between ProtEx and BLAST.

| Task | Metric | Null Stdev. | Observed Diff | $p$-value |
|------|--------|-------------|---------------|-----------|
| Random EC | Micro F1 | 0.0002 | 0.003 | 1.32e-24 |
| Clustered EC | Micro F1 | 0.0002 | 0.008 | 1.53e-59 |
| PDB EC | Protein-centric F1 | 0.0031 | 0.009 | 6.17e-3 |

Table 10: Variance between fine-tuning runs.

| Task | Metric | Fine-tuning Stdev. |
|------|--------|--------------------|
| PDB EC | Protein-centric F1 | 0.0006 |
| NEW-392 | Weighted AUC | 0.0003 |
| Price-149 | Weighted AUC | 0.0009 |

## C.2 MODEL ARCHITECTURES AND SCALING

The computational cost of self-attention in a standard Transformer scales quadratically with input length. To more efficiently encode exemplars, we studied the Fusion-in-Decoder (FiD) approach introduced by Izacard & Grave (2021). We apply this approach to encode the query and each exemplar in a separate encoder, effectively masking attention between exemplars. Notably, this model variant consists of the same set of parameters and can be initialized from the same pre-trained checkpoint. We compare the performance of the standard and FiD architecture for Small (60M) models in Table 11.

The FiD architecture performs only slightly worse than the standard Transformer, indicating this may be one path towards more efficiently encoding a larger number of exemplars. On the other hand, the drop in performance suggests there is value in attention across exemplar sequences.

Another potential alternative would be a specialized architecture such as MSA Transformer (Rao et al., 2021). This would require a couple of modifications to the MSA Transformer architecture. First, MSA Transformer would need to be adapted to include functional labels along with unlabeled sequences. Second, the architecture requires that all sequences are aligned as a preprocessing step. Intuitively, our pre-training and fine-tuning procedures are designed to teach the model to implicitly align sequences without relying on heuristic alignments. Notably, the main difference between MSA Transformer and a standard Transformer is the more restricted attention operations allowed in MSA Transformer. As we have shown, using a Fusion-in-Decoder Transformer, which has a more restricted attention mechanism, leads to a modest drop in performance. Therefore, this would be a concern for any architecture that similarly restricts the attention mechanism such as MSA Transformer. Regardless, such specialized architectures could be a path to explore for future work.

Table 11: **Model Architecture Comparisons.** We report F1 on Clustered EC and GO development splits for standard Transformer vs. Fusion-in-Decoder (FiD) for Small models

| Architecture | EC | GO |
|--------------|-----|-----|
| Standard Transformer | **0.958** | **0.845** |
| FiD Transformer | 0.953 | 0.842 |

We also compare Small vs. Base size models in Table 12. Given the full development set is quite large (approximately 180,000 examples), we performed both these ablations on a random 10% subset.

## C.3 ADDITIONAL RESULTS ON EC PREDICTION

### C.3.1 ABLATING LABELS IN INPUT

We evaluate how much the model's performance depends on being able to condition on the candidate label as input. Table 13 shows that the model achieves similar performance with and without the candidate label, indicating that the model is indeed conditioning on the exemplars. Given the full

Table 12: **Model Size Comparisons.** We report F1 on Clustered EC and GO development splits for Small and Base sized Transformers.

| Size | EC | GO |
|---|---|---|
| Base (220M) | **0.959** | **0.848** |
| Small (60M) | 0.958 | 0.845 |

development set is quite large (approximately 180,000 examples), we performed this ablation on a random 10% subset.

Table 13: **Label Ablation.** Comparing whether we include the label being predicted in the input as shown in Figure 1, or not. Results are max F1 on the clustered EC and GO development set for Small models. Model performance is only slightly lower without access to the label.

| Method | EC | GO |
|---|---|---|
| ProtEx-Small (with input label) | **0.958** | **0.845** |
| ProtEx-Small (without input label) | 0.957 | 0.843 |

### C.3.2 COMPARING RETRIEVAL STRATEGIES

We compared the BLAST and per-class retrieval strategies for EC prediction on the clustered split in Table 14. Using BLAST to filter the number of classes gives considerably stronger performance on EC prediction, likely because the BLAST performance is quite high. As a result, we used the BLAST retrieval strategy for all the EC and GO results. As with Table 13, given the full development set is quite large (approximately 180,000 examples), we performed this ablation on a random 10% subset.

Table 14: **Retrieval Strategy Comparison**: Comparing the BLAST and per class retrieval approaches on EC prediction (clustered development split).

| Method | EC |
|---|---|
| ProtEx (BLAST retrieval) | **0.959** |
| ProtEx (Per Class retrieval) | 0.929 |

### C.3.3 NEW-392 AND PRICE-149 RESULTS

The weighted AUC metric proposed by Yu et al. (2023b) averages F1 scores over classes based on their representation in the test set. Especially since the NEW-392 and Price-149 test sets only include a small subset of classes, this metric tends to emphasize higher recall and lower precision relative to more standard metrics such as micro-averaged F1. Therefore, we also report the maximum micro-averaged F1 scores for NEW-392 and Price-149 for ProtEx and BLAST in Table 15.

Table 15: Maximum Micro F1 scores for EC prediction for NEW-392 and Price-149 evaluation sets.

| Method | NEW-392 | Price-149 |
|---|---|---|
| BLAST | 0.593 | 0.391 |
| ProtEx | **0.612** | **0.441** |

### C.4 ADDITIONAL RESULTS ON PFAM

### C.4.1 ANALYZING EFFECT OF CLASS FILTERING

Unlike for the EC and GO tasks, in Pfam we do not use a candidate label generator and consider all potential classes for each query sequence. We made this decision based on the following analysis.

We used `PairwiseAligner` (as in Appendix B.5) to select the single closest exemplar per candidate label for a random subset of 1000 sequences in the development set which gives $|\mathcal{L}|$ exemplars for each sequence. We can then restrict the number of candidate labels to $K$ by taking the corresponding classes for the closest $K$ exemplars to the query sequence in this selected set. Table 16 shows the results for various values of $K$, showing that it is beneficial to consider a large number of classes.

Table 16: **Analysis of Class Filtering for Pfam**: Table showing how filtering by a homology based approach (`PairwiseAligner`) reduces the accuracy ceiling.

| Number of Candidate Labels | Accuracy Ceiling |
| --- | --- |
| 10 | 82.3 |
| 50 | 87.8 |
| 100 | 88.7 |
| 500 | 93.4 |
| 1000 | 95.3 |
| 2000 | 96.2 |
| 5000 | 98.2 |
| 17929 | 100 |

### C.4.2 NO EXEMPLAR ABLATION

We experiment with ablations for Pfam that remove exemplars. The first is to finetune the model following the procedure as our other results for ProtEx *i.e.* to generate per-class binary predictions, but with no exemplars. The second strategy is to finetune our pretrained checkpoint to directly predict the label string from the sequence.

As shown in Table 17, we find that the first approach performs poorly compared to the second. Upon further analysis, we believe the reason for this is the large number of classes (17,929) in Pfam. We hypothesize that, without exemplars, the model does not learn to effectively discriminate between the positive class and all competing classes when trained using binary supervision, which requires sampling of negative classes to avoid class imbalance.

Table 17: **No Exemplar Ablation Comparison**: Comparing different no exemplar approaches for Pfam seed.

| Method | Family Accuracy (Dev) |
| --- | --- |
| No Exemplar Binary Prediction | 40.3 |
| No Exemplar Label String Prediction | 74.7 |

### C.4.3 STRATIFIED PERFORMANCE

We show family accuracy stratified by sequence similarity in Figure 7. This shows that our approach consistently performs well across sequences with low similarity to the closest sequence in the training data. We also show the stratified performance by lifted clan accuracy in Figure 8 and Figure 9 that shows similar trends to Figure 7 and Figure 5.

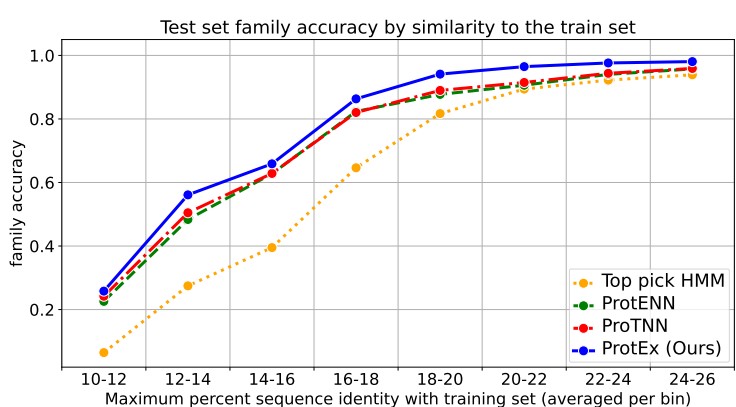

Figure 7: Pfam family accuracy stratified by sequence similarity.

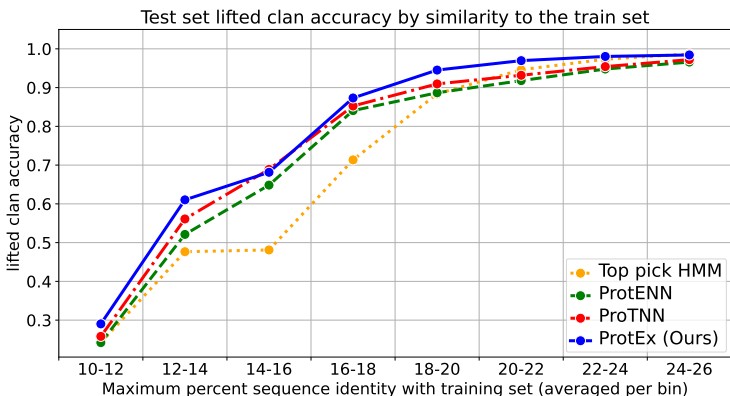

Figure 8: Pfam lifted clan accuracy stratified by sequence similarity.

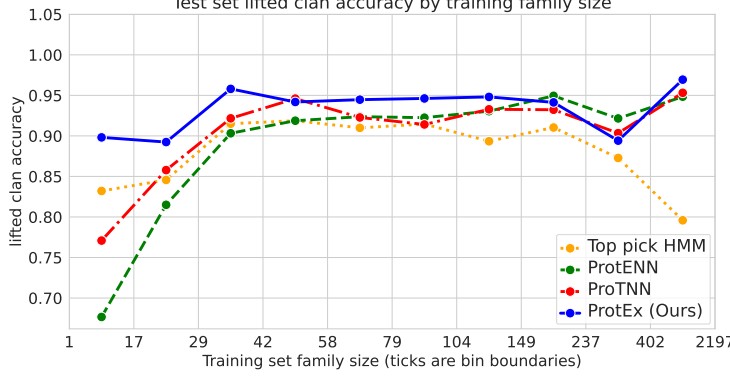

Figure 9: Pfam lifted clan accuracy stratified by number of training examples per class.

