# OpenReview forum: "ProtEx: A Retrieval-Augmented Approach for Protein Function Prediction"
_ICLR.cc/2025/Conference — Submitted to ICLR 2025_

### Official Review · Reviewer_nQPw · 2024-11-03

**Soundness:** 3
**Presentation:** 2
**Contribution:** 2
**Rating:** 6
**Confidence:** 3

**Summary:**

This paper presents ProtEx, a retrieval-augmented model designed to enhance protein function prediction by leveraging exemplar sequences from a reference database. By conditioning predictions on retrieved exemplars, the model aims to improve accuracy and generalization, especially for underrepresented classes. ProtEx is applied to various functional annotation tasks, such as predicting EC numbers, GO terms, and Pfam families, where it reportedly achieves state-of-the-art performance. The proposed approach includes a unique multi-sequence pretraining task and a targeted fine-tuning strategy to maximize the benefit of exemplar retrieval.

**Strengths:**

- The integration of exemplars into protein function prediction is a novel direction that could address challenges with generalization and robustness, particularly for underrepresented classes in protein datasets. This retrieval-augmented approach could be valuable in instances where limited data is available for specific protein functions during training.
-  ProtEx is tested across diverse protein function prediction tasks, including EC, GO, and Pfam annotations. This extensive evaluation provides insights into the method’s versatility and potential applicability to various protein annotation domains.
- The authors provide proper ablation studies and analyses showing ProtEx’s capability to improve performance on classes with limited representation in the training data, a common challenge in protein function prediction.
- The research is well motivated and of good biological significance.

**Weaknesses:**

- While retrieval-augmented models are relatively new in the protein domain, the approach is already widely used in other areas, such as natural language processing. For broader readers,  the authors may consider use a few illustrative figures or tables to justify the significance of retrieval-augmentation for protein function prediction. (I am not questioning this, I believe in the importance of similarity-based prediction for protein, and think this will further enhance the quality of manuscript for readers who are not familiar)
- The performance of ProtEx heavily depends on the quality and comprehensiveness of the reference database. In cases where exemplars are sparse or unavailable for certain protein functions, the retrieval mechanism may underperform, potentially limiting the model’s utility in practical applications with limited exemplar coverage. Only BLAST is tested for similarity search, which may weaken the soundness of this study.
- Although ProtEx aims to improve generalization, its effectiveness for predicting completely novel functions is not well-supported in the experiments. The evaluation focuses primarily on tasks with annotated classes, leaving uncertainty about the model’s robustness in generalizing to truly unseen or hypothetical functions, a critical aspect of protein function prediction. As a matter of fact, a lot of EC classes, for example, are labeled by expert largely based on the sequence similarity and that explains why the BLAST baseline alone yields already good results.
- The paper primarily compares ProtEx against few retrieval-based and BLAST baselines but does not provide extensive comparison with other models (especially, **under their data train-test split for comparison**) that may be equally effective for protein function prediction. This makes it difficult to assess the value of ProtEx. For example, in one of the mentioned papers, ProtIR, I saw the authors there have benchmarked across different types of models on specific data splits and their BLAST results have a distinguishable mismatch with the one shown in this paper. Add such comparison can improve the soundness of this paper.

**Questions:**

- Have you explored using other sequence-based / structure-based alignment or search engines, potentially with different retrieval mechanisms, to assess if ProtEx’s performance varies based on the retrieval engine? For example, foldseek, mmseqs, and even neural-based models. Such experiments could clarify the model’s dependency on specific retrieval tools.
- Have you considered applying ProtEx (as an encoder) as an augmentation layer on top of existing protein embeddings rather than directly on raw sequences? Testing ProtEx in this way might provide insights into whether it could benefit from the structural and functional information encoded in precomputed protein embeddings.
- Could you provide an analysis of how ProtEx performs when fewer exemplar sequences are available, or even when limited to a small number of retrieved sequences? For example the orphan protein. This could help in understanding ProtEx’s robustness and effectiveness when retrieval options are sparse.

---

> ### Author Response · Authors · 2024-11-16
>
> Thank you for reviewing our work!
>
> > "For broader readers, the authors may consider use a few illustrative figures or tables to justify the significance of retrieval-augmentation for protein function prediction"
>
> Thank you, this is a great idea, and we indeed hope to make the paper interesting and accessible to readers who may have some familiarity with retrieval-augmented models (e.g. from NLP) but not protein function prediction, and vice versa. We will look to extend Figure 1 to more clearly contrast our approach with standard methods for protein function prediction that do not condition predictions on retrieved exemplars in our camera ready version.
>
> > "The performance of ProtEx heavily depends on the quality and comprehensiveness of the reference database. In cases where exemplars are sparse or unavailable for certain protein functions, the retrieval mechanism may underperform, potentially limiting the model’s utility in practical applications with limited exemplar coverage. Only BLAST is tested for similarity search, which may weaken the soundness of this study."
>
> Please see our new results below!
>
> > "Although ProtEx aims to improve generalization, its effectiveness for predicting completely novel functions is not well-supported in the experiments. The evaluation focuses primarily on tasks with annotated classes, leaving uncertainty about the model’s robustness in generalizing to truly unseen or hypothetical functions, a critical aspect of protein function prediction. As a matter of fact, a lot of EC classes, for example, are labeled by expert largely based on the sequence similarity and that explains why the BLAST baseline alone yields already good results."
>
> We agree with your point in general, as predicting completely novel functions is an interesting problem but beyond the scope of this paper. However, we argue that more accurately predicting functional labels within existing ontologies such EC numbers, GO terms, and Pfam domains can still be useful, and indeed is a widely studied problem. Furthermore, we have shown that ProtEx can generalize to new functions not seen at training time (Table 6). This is interesting because new labels are frequently added to these ontologies, and ProtEx can adapt to these new labels without any re-training (although it does require at least 1 positive exemplar sequence for the new label).
>
> We also evaluated ProtEx on settings where the sequence similarity between test and train sequences is limited. Table 2 shows performance on clustered splits that restrict similarity between train and test sequences, Table 3 shows performance on specifically curated challenge sets, Table 4 partitions performance by sequence similarity, and Table 5 also shows performance on a clustered split that restricts similarity between train and test sequences. In all cases ProtEx demonstrates significant improvements over methods from prior work.
>
> (Response continued in next comment.)

---

> > ### Author Response · Authors · 2024-11-16
> >
> > > The paper primarily compares ProtEx against few retrieval-based and BLAST baselines but does not provide extensive comparison with other models (especially, under their data train-test split for comparison) that may be equally effective for protein function prediction. This makes it difficult to assess the value of ProtEx. For example, in one of the mentioned papers, ProtIR, I saw the authors there have benchmarked across different types of models on specific data splits […]. Add such comparison can improve the soundness of this paper.
> >
> > We think there may be some confusion here, which we apologize for and will aim to make clearer in our paper. __All of our experiments use standardized train and test splits proposed by prior work and all comparisons across models are evaluated using the same train and test splits. We compare with (and outperform) 20+ neural methods across the tasks that we study, including the strongest ones from prior work.__ Our comparisons in the paper include both retrieval-based methods *and* methods that directly predict labels from representations of sequences and/or structures.
> >
> > For example, in our submission, we report results for ProtEx on the widely studied PDB-based EC task that the ProtIR paper studied in Table 4. For brevity, Table 4 only included the strongest approaches from prior work. However, please see this more complete table of comparisons with other neural models evaluated on the same train and test split, which we will include in the appendix for the camera ready version.
> >
> > | Method |  Result Source | Max F1 |
> > | --- | --- | --- |
> > | CNN (Shanehsazzedeh et al., 2020) | [3] | 0.545 |
> > | ResNet (Rao et al., 2019) | [3] | 0.605 |
> > | LSTM (Rao et al., 2019) | [3] | 0.425 |
> > | Transformer (Rao et al., 2019) | [3] | 0.238 |
> > | GCN (Kipf et al., 2017) | [3] | 0.320 |
> > | GAT (Velickovic et al., 2018) | [3] | 0.368 |
> > | GVP (Jing et al., 2021) | [3] | 0.489 |
> > | GraphQA (Baldassarre et al., 2021) | [3] | 0.509 |
> > | New IEConv (Hermosilla et al., 2022) | [3] | 0.735 |
> > | DeepFRI [1] | [1] | 0.631 |
> > | ESM-1b (Rives et al., 2021) | [3] | 0.864 |
> > | ProtBERT-BFD (Elnaggar et al., 2021) | [3] | 0.838 |
> > | LM-GVP [2] | [3] | 0.664 |
> > | GearNet-MVC [3] | [3]  | 0.874 |
> > | ESM+GearNet | [4] | 0.890 |
> > | ESM+GVP  | [4] | 0.881 |
> > | ESM+CDConv  | [4] | 0.880 |
> > | TM-Vec (Hamamsy et al., 2022) | [5] | 0.817 |
> > | ProtIR [5] (w/ GearNet) | [5] | 0.881 |
> > | ProtIR [5] (w/ CDConv) | [5] | 0.885 |
> > | ESM-2 (Lin et al., 2023) (650M variant) | [5] | 0.877 |
> > | PromptProtein [6] | [6] | 0.888 |
> > | OntoProtein (Zhang et al., 2022) | [7]  | 0.841 |
> > | ProtST [7] (w/ ProtBERT) | [7] | 0.856 |
> > | ProtST [7] (w/ ESM-1b) | [7] | 0.878 |
> > | ProtST [7] (w/ ESM-2) | [7] | 0.878 |
> > | ESM-2 (classifier head) | [8] | 0.892 |
> > | PST [9] | [9] | 0.899 |
> > | ProtEx | | __0.909__ |
> >
> > [1] “Structure-based protein function prediction using graph convolutional networks” Gligorijevic et al. (2021) - https://www.nature.com/articles/s41467-021-23303-9
> >
> > [2] “LM-GVP: an extensible sequence and structure informed deep learning framework for protein property prediction” Wang et al. (2022) - https://www.nature.com/articles/s41598-022-10775-y
> >
> > [3] “Protein Representation Learning by Geometric Structure Pretraining” Zhang et al. (2023) - https://openreview.net/forum?id=to3qCB3tOh9
> >
> > [4] “A Systematic Study of Joint Representation Learning on Protein Sequences and Structures” Zhang et al. (2023) - https://arxiv.org/abs/2303.06275
> >
> > [5] “ProtIR: Iterative Refinement between Retrievers and Predictors for Protein Function Annotation” Zhang et al. (2024) - https://arxiv.org/abs/2402.07955
> >
> > [6] “Multi-level Protein Structure Pre-training via Prompt Learning” Wang et al. (2023) - https://openreview.net/forum?id=XGagtiJ8XC
> >
> > [7] “ProtST: Multi-Modality Learning of Protein Sequences and Biomedical Texts” Xu et al. (2023) - https://arxiv.org/abs/2301.12040
> >
> > [8] “Endowing Protein Language Models with Structural Knowledge” Chen et al. (2024) - https://arxiv.org/abs/2401.14819v1
> >
> > Finally, we applaud Zhang et al. for their efforts to establish a large set of benchmark results on this standardized split, and our comparisons benefit from their work. However, we note that we actually compare ProtEx with *more* previous approaches than ProtIR compared with, as we also study the SwissProt-based splits of Sanderson et al. and the Pfam task, enabling comparisons with strong neural models from prior work such as ProteInfer (Sanderson et al. 2023), ProtENN (Bileschi et al. 2022), ProtTNN (Dohan et al. 2021), and ProtNLM (Gane et al. 2022). ProtEx also outperforms ProtNote (Char et al., 2024), a method described in a preprint from after the ICLR submission date, on the SwissProt-based split.
> >
> > (Response continued in next comment.)

---

> > > ### Author Response · Authors · 2024-11-16
> > >
> > > > "[ProtIR’s] BLAST results have a distinguishable mismatch with the one shown in [the ProtIR] paper."
> > >
> > > We tried to address this with a note in section 4.2 “In some cases, our BLAST results are stronger than those reported by prior work due to this thresholding.”, but should have been more specific. We were also initially curious why our BLAST results differed from those reported by ProtIR. Prior to submission, we contacted the ProtIR authors and they were kind enough to provide us with their evaluation scripts and to verify their exact training data preprocessing. We validated that our experimental setting exactly matched theirs. We also aligned on BLAST version and hyperparameters. The small differences in BLAST results can be attributed to differences in aggregation. We follow Sanderson et al. and impute the label from the top-1 result if it exceeds a score threshold, while the aggregation of ProtIR is more complex (described in equation 2 of their paper) and inspired by Kernel-based methods. It seems in this case, the simpler aggregation strategy can be more effective. Our open-source code (attached to submission) provides end-to-end instructions for reproducing the BLAST results we report on the standard PDB EC split.
> > >
> > > > "Have you explored using other sequence-based / structure-based alignment or search engines, potentially with different retrieval mechanisms, to assess if ProtEx’s performance varies based on the retrieval engine? For example, foldseek, mmseqs, and even neural-based models. Such experiments could clarify the model’s dependency on specific retrieval tools."
> > >
> > > Based on your suggestion, we have analyzed the performance of ProtEx using alternative retrieval strategies to support the claim that the overall approach is not necessarily specific to using BLAST. To ensure a timely response, we focused on the PDB-based EC split, as this is one of the smaller datasets that we studied. This split also has PDB structures available so that we can compare structure-based retrievers such as Foldseek. The table below shows the end-to-end performance of ProtEx when using Foldseek or MMSeqs2 instead of BLAST (using standard configurations for those retrievers).
> > >
> > > | ProtEx Variant | Max F1 |
> > > | --- | --- |
> > > | No exemplars | 0.849 |
> > > | BLAST exemplars | 0.909 |
> > > | Foldseek exemplars | 0.908 |
> > > | MMSeqs2 exemplars | 0.899 |
> > >
> > > We also include results for these retrievers on their own, imputing the labels from the top retrieved sequence if it exceeds a score threshold (as we did for our BLAST results):
> > >
> > > | Retriever | Max F1 |
> > > | --- | --- |
> > > | BLAST | 0.900 |
> > > | Foldseek | 0.898 |
> > > | MMSeqs2 | 0.890 |
> > >
> > > We see that ProtEx can still outperform the prior SOTA on this task using Foldseek instead of BLAST. The end-to-end results with ProtEx correlate with the strength of the retriever in isolation, but in all cases ProtEx outperforms using the retriever in isolation. It is perhaps notable that BLAST performs comparably to Foldseek, given that Foldseek leverages PDB structures (although this aligns with trends from prior work). MMSeqs2 performs slightly worse than BLAST, which is to be expected: it was designed to be faster than BLAST but can be less sensitive. Regardless, using MMSeqs2 exemplars can still lead to competitive performance.
> > >
> > > > "Have you considered applying ProtEx (as an encoder) as an augmentation layer on top of existing protein embeddings rather than directly on raw sequences? Testing ProtEx in this way might provide insights into whether it could benefit from the structural and functional information encoded in precomputed protein embeddings."
> > >
> > > In general, we agree that it would be interesting to consider how to incorporate existing protein encoders or embeddings, especially those that capture information not currently included in the ProtEx encoder (e.g. structural information). We hope that the key findings of ProtEx would be complementary to such a direction. However, we considered this out of scope for the current study so that we could focus our exploration on aspects of the pre-training and fine-tuning procedures most relevant to improving the ability to condition predictions on retrieved exemplars.
> > >
> > > (Response continued in next comment.)

---

> ### Author Response · Authors · 2024-11-16
>
> > "Could you provide an analysis of how ProtEx performs when fewer exemplar sequences are available, or even when limited to a small number of retrieved sequences? For example the orphan protein. This could help in understanding ProtEx’s robustness and effectiveness when retrieval options are sparse."
>
> We agree this is a crucial question. While this can be a challenging case for any ML-based method, we find that the relative performance of ProtEx is actually strongest in cases where fewer reference examples are available for a given class.
>
> In the original submission, we provided analysis of how well ProtEx performs when few exemplars are available for a given class. Per Figure 5, the relative performance of ProtEx is actually strongest for the bucket representing test sequences belonging to classes with only 1-17 reference sequences.
>
> To further analyze this, we have also performed a similar analysis for EC-based prediction below, focusing on the clustered EC split. We analyze performance when we restrict the computation of the max micro-averaged F1 score to classes in the dev set with less than a given number of training sequences. Notably, conditioning on exemplars is particularly effective for these rare classes relative to our baseline without exemplars and ProteInfer (from Sanderson et al., 2023), highlighting a strength of a retrieval-augmented approach such as ProtEx.
>
> | | Max F1 | | | | | |
> | --- | --- | --- | --- | --- | --- | --- |
> | # of training sequences per class | 1 | 1-2 | 1-4 | 1-8 | 1-16 | All Classes |
> | # of classes | 903 | 1312 | 1779 | 2180 | 2454 | 3411 |
> | ProtEx | __0.700__ | __0.668__ | __0.706__ | __0.745__ | __0.789__ | __0.959__ |
> | ProtEx (no exemplars) | 0.157 | 0.322 | 0.474 | 0.552 | 0.646 | 0.944 |
> | ProteInfer | 0.412 | 0.437 | 0.494 | 0.538 | 0.616 | 0.910 |

---

> > ### Author Response · Authors · 2024-11-20
> >
> > We have edited the above comment to also include ProteInfer results in the table evaluating performance for rare classes for EC number prediction.

---

> ### Comment · Reviewer_nQPw · 2024-11-21
> **Response to the authors' rebuttal**
>
> Thanks for the detailed answer for addressing my question. I hope the discussion did help improve the quality of manuscript :) I have adjusted my score to 6.

---

> > ### Author Response · Authors · 2024-11-21
> >
> > Thank you! We really appreciate the discussion and helpful suggestions!

---

### Official Review · Reviewer_JGPY · 2024-11-04

**Soundness:** 2
**Presentation:** 2
**Contribution:** 2
**Rating:** 5
**Confidence:** 3

**Summary:**

The paper introduces ProtEx, a retrieval-augmented approach for protein function prediction, emphasizing the importance of conditioning predictions on exemplar sequences, particularly for less represented classes and sequences in the training data. The approach is novel in integrating retrieval into protein prediction tasks, improving generalization across diverse protein classes, and effectively addressing the data scarcity for certain sequences.

**Strengths:**

1. The empirical performance for ProtEx outperforms other reported baselines on essential protein function prediction tasks.
2. The introduction of a retrieval-augmented approach to protein function prediction is innovative, particularly the use of exemplar conditioning which helps in handling underrepresented data effectively.

**Weaknesses:**

1. The increased computational complexity due to the conditioning on labels is a drawback, particularly if scalability or application to datasets with larger classes is considered.

2. Some experimental details are not clear. See questions.

**Questions:**

1. In Sec. 3, the proposed method is conditioned on label and labels are enumerated. What's the advantage for the conditioning?

2. Since ProtEx uses BLAST to produce the pre-training and fine-tuning datasets for exampler, it would be reasonable for ProtEx to outperform BLAST. Could the integration of BLAST results with ProtEx be considered to enhance performance? If not, maybe ProtEx totally absorbs the knowledge from BLAST

3. Also, when not using examplars for fine-tuning, i.e., not using BLAST for help, ProtEx perform even much worse than BLAST.  In this sense, the largest constribution for ProtEx seems to be incorporating BLAST for downstream tasks, instead of its pre-training or fine-tuning strategies. How do the authors justify the innovation in their approach given the above thinking?

4. In Tab. 6, if the candidate labels are not incorporated, how the prediction is done, since the model cannot take it as input?

---

> ### Author Response · Authors · 2024-11-16
>
> Thank you for reviewing our work!
>
> > "In Sec. 3, the proposed method is conditioned on label and labels are enumerated. What's the advantage for the conditioning?"
>
> We are not sure whether you are asking about the advantage of conditioning the selection of positive and negative exemplars based on a label, or the advantage of including the label as an explicit input to the model. So we will address both here.
>
> Overall, the main advantage of making a prediction specific to a given label is so that the positive and negative exemplars can be selected based on a particular label. Some of the rationale for this is discussed in the beginning of section 3.1. It is important to note that because our model only has a limited context window, we can only condition each prediction on a limited number of exemplars. Selecting these exemplars conditioned on a label lets us focus each prediction on the exemplars that are most relevant towards understanding the class boundary of a specific label, as illustrated in Figure 1. For the same query sequence, but another label, we can then select a different set of exemplars for understanding the class boundary.  Otherwise, if we selected a single set of exemplars per sequence, most classes would likely not be well represented in this set.
>
> Whether we include the label itself as an input to the model (as shown in Figure 1) or not does not have a significant impact on performance empirically (see the ablation shown in Table 13 in the appendix). This is an interesting finding, because without the label explicitly represented, the model must rely entirely on the information provided by the positive and negative exemplars to determine the relevance of the given class. This ability is further validated by the results in Table 6.
>
> However, including the label as input allows for a clearer ablation of the impact of conditioning on the exemplars. Understanding the impact of conditioning on exemplars is central to the paper and this ablation is included in all of our main result tables. Without the label *or* exemplars, the prediction task would be underspecified.
>
> > "In Tab. 6, if the candidate labels are not incorporated, how the prediction is done, since the model cannot take it as input?"
>
> This is probably clearest to explain by considering the example given in Figure 1. Even without explicitly considering the candidate label (e.g. EC:1.2.3.4), it is still possible to make a binary judgment based on whether the input sequence is more similar to the positive or the negative exemplars.
>
> > "Since ProtEx uses BLAST to produce the pre-training and fine-tuning datasets for exampler, it would be reasonable for ProtEx to outperform BLAST. Could the integration of BLAST results with ProtEx be considered to enhance performance? If not, maybe ProtEx totally absorbs the knowledge from BLAST"
>
> We note that it is non-trivial to outperform BLAST on the tasks that we study. On many of the tasks, BLAST outperforms the previous best results from neural models. Given this observation, the core motivation for ProtEx is to combine the relative strengths of pre-trained neural networks with similarity-search based methods such as BLAST.
>
> Notably, it would be relatively trivial for a retrieval-augmented model to *match* the BLAST baseline. The model could simply learn the heuristic of copying the prediction associated with most similar sequence retrieved by BLAST. What is interesting about ProtEx is that it consistently *outperforms* BLAST. Further, Figure 4 demonstrates that ProtEx can outperform BLAST at *all* points of their respective PR curves. Therefore, ProtEx is not only recapitulating BLAST. ProtEx can aggregate information across multiple exemplars using a learned non-linear model, transfer knowledge learned during pre-training, and learn task-specific notions of similarity between query and exemplar sequences during fine-tuning. As we show below, our proposed pre-training and fine-tuning procedures are critical for ProtEx to outperform BLAST.
>
> Moreover, we note that on the Pfam task (Table 5), BLAST performs poorly, perhaps due to the construction of the split which ensures that the sequences in the test set have at most 25% sequence identity to the train set). ProtEx, on the other hand, achieves SOTA performance, indicating the robustness of our approach.
>
> Finally, while we primarily focused on BLAST as the source of exemplars in this paper, as it is a widely used and understood tool, our method is not specific to BLAST. We offer some new results using Foldseek and MMSeqs2 as alternatives to BLAST and still demonstrate strong performance (please see our response to Reviewer nQPw).
>
> (Response continued in next comment.)

---

> > ### Author Response · Authors · 2024-11-16
> >
> > > "Also, when not using examplars for fine-tuning, i.e., not using BLAST for help, ProtEx perform even much worse than BLAST. In this sense, the largest constribution for ProtEx seems to be incorporating BLAST for downstream tasks, instead of its pre-training or fine-tuning strategies. How do the authors justify the innovation in their approach given the above thinking?"
> >
> > The core finding of our paper is that conditioning predictions on retrieved exemplars can lead to significant improvements for protein function prediction. Our proposed pre-training and fine-tuning recipes are important for achieving this result.
> >
> > To better understand the contribution of the pre-training and fine-tuning strategies we have completed a more complete set of ablations on the clustered EC task. We compare our proposed multi-sequence pre-training with the more standard single-sequence pre-training. We also compare our proposed recipe to fine-tune models with exemplars compared to fine-tuning without exemplars. The results below are on the dev set of the clustered EC split, building on the results in Table 7.
> >
> > | Multi-sequence pre-training | Fine-tuning with Exemplars | Inference with Exemplars | Max F1 |
> > |---|---|---|---|
> > | No | No | No | 0.923 |
> > | No | Yes | Yes | 0.952 |
> > | Yes | No | Yes | 0.459 |
> > | Yes | Yes | Yes | __0.958__ |
> >
> > Therefore, we see that our proposed pre-training and fine-tuning procedures are essential to fully leverage exemplars at inference time. Without including exemplars during fine-tuning, the model achieves extremely poor performance if exemplars are included at inference time, due to the shift in input format. While the effect of pre-training on single sequences vs. multi-sequences is less substantial, it is still significant. We also offer more analysis of how aspects of the pre-training and fine-tuning procedures affect model performance in the submission, e.g. Tables 7 and Figure 6.

---

> > > ### Author Response · Authors · 2024-11-25
> > >
> > > Dear reviewer JGPY,
> > >
> > > Thank you again for your time and effort in reviewing our paper and responses! As the discussion deadline is approaching, we would like to follow up to see if the response addresses your concerns or if you have any further questions or recommendations.

---

### Official Review · Reviewer_opKf · 2024-11-04

**Soundness:** 3
**Presentation:** 3
**Contribution:** 3
**Rating:** 6
**Confidence:** 3

**Summary:**

The authors propose ProtEx, a novel approach to use retrieval-augmentation to enhance protein language models in protein function prediction tasks. The retrieval part is done by a devoted language model to determine the similarity between two protein sequences. The information of retrieved sequences are then integrated into query sequence to determine the prediction labels. The author conduct extensive experiments to show the validity of their approach, and conduct several ablation study.

**Strengths:**

1. The paper is well-written and clearly structured. The details are clearly explained in training and experiments part.
2. The idea of retrieval-augmentation for protein function prediction is nice, though not novel. However, the paper is devoted to its methodology part, and the experiments show that their method outperforms current SOTA.
3. I really like the various analysis and ablation studies in experiment part. They are quite convincing and helpful to the community.

**Weaknesses:**

1. It is to be discussed whether the paper is novel enough for a machine learning conference. pro: the task of protein function prediction with retrieval-augmentation is not well-explored today, and the paper provides meaningful discussion. con: the methodology itself is not novel. I can see that some parts are well designed, like the old-BERT-style pretraining loss, but I don't think they are contributions in a machine learning conference.

**Questions:**

The paper is well written to its claimed scope. I have no further questions.

---

> ### Author Response · Authors · 2024-11-16
>
> Thank you for reviewing our work!
>
> > "It is to be discussed whether the paper is novel enough for a machine learning conference. pro: the task of protein function prediction with retrieval-augmentation is not well-explored today, and the paper provides meaningful discussion. con: the methodology itself is not novel."
>
> We believe the simplicity of our approach is an advantage, as we show it outperforms more complex approaches from prior work (e.g. Table 4). Our method is carefully designed and we provide ablations showing the value of each component of our method (e.g. the novel pretraining strategy).
>
> Novelty of the methods aside, we believe our findings are likely to be of significant interest to at least two groups within the ICLR community:
>
> 1. __Researchers interested in retrieval-augmented modeling methods and applications.__ Protein function prediction has many properties (rare classes, sequences far from the training data, large label space) that we show are very amenable for retrieval-augmented methods and also offer new challenges for researchers in this area. Identifying an area where such methods can have such a significant impact will likely be interesting to this community. Thus, we believe our results can help introduce methodological researchers to an emerging application area.
> 2. __Researchers and practitioners interested in improving the ability to characterize proteins.__ We believe this is a very impactful application of ML, and we note that some of the prior work we compare with has been published in ML venues, e.g. ProtST (ICML 2023), PromptProtein (ICLR 2023), GearNet (ICLR 2023). We believe that our key finding that retrieval-augmented methods can enable significant improvements over the current state-of-the-art is novel and well supported by our experiments and analysis, and can inspire future work.

---

### Author Response · Authors · 2024-11-25

We thank all of the reviewers for their time and effort in reviewing our submission! We responded to each reviewer separately, but wanted to highlight the additional experiments/comparisons that we included in the responses that will be included in the appendix for the camera ready version.

* Additional ablations of pre-training and fine-tuning with and without exemplars (Reviewer JGPY): We provide experiments in our response showing how our pretraining and finetuning strategy is critical to effectively utilize exemplars to achieve SOTA performance.

* Additional retrievers (Reviewers JGPY, nQPw): We include experiments in our response to Reviewer nQPw showing that our method still performs well with other retrievers (FoldSeek, MMSeqs2) indicating that ProtEx is not BLAST-specific and in each case, outperforms the retriever in isolation.

* Additional analysis results for small classes (Reviewer nQPw): On the clustered EC split, we provide additional analysis demonstrating the strong relative performance of ProtEx when few exemplars are available for a given class. Figure 5 in the main paper provides further evidence of this claim.

* Expanded Table 4 (Reviewer nQPw):  For brevity, Table 4 (PDB-based EC task) only includes the strongest approaches from prior work. In our response, we include a more complete table of comparisons with other neural models that have been evaluated on this widely studied train and test split.

---

### Meta-Review · Area_Chair_NXgC · 2024-12-21

**Metareview:**

The paper considers the task of protein function prediction based on its sequence. A retrieval-augmented approach is introduced where predictions are conditioned on exemplars that are retrieved from a database, aiming to improve accuracy and generalization, especially targeting underrepresented classes.

The AC and reviewers all found that the paper is well written and that it tackles an important problem. The use of exemplar based conditioning is intuitive and interesting. Extensive evaluation and ablation studies were conducted.

While the approach achieves strong empirical results, the computational overhead due to the conditioning is quite significant. In addition, the novelty of the methodology is somewhat limited, and retrieval-based approaches for protein function prediction have already been proposed in the literature that  make the case that such approaches are useful in improving the performance of predictive approaches.

**Additional Comments On Reviewer Discussion:**

The reviewers raised points related to limited methodological novelty, increased computational complexity due to conditioning, lack of comparison against closely related approaches, comparison with BLAST, reliance on the quality of the database and the need to test on a variety of retrieval engines.
The authors have provided valuable clarifying comments, additional evaluation results and analysis to address those concerns. However the novelty remains limited, methodologically and in view of previously proposed retrieval-based approaches proposed for the same task. Also, the computational overhead of the exemplar conditioning is non-neglectible and it would be beneficial to report the training time of various approaches as done e.g. in in the ProtIR paper.

---

### Decision · Program_Chairs · 2025-01-22

Reject